# Influence of Toroidal Flow on Stationary Density of Collisionless Plasmas

**Elias Laribi [1], Shun Ogawa [2] , Guilhem Dif-Pradalier [1], Alexei Vasiliev [3], Xavier Garbet [1] and Xavier Leoncini [4,\***

[1]  CEA, IRFM, F-13108 St. Paul-lez-Durance CEDEX, France; elias.laribi@ens-lyon.fr (E.L.); guilhem.dif-pradalier@cea.fr (G.D.-P.); xavier.garbet@cea.fr (X.G.)

[2]  Laboratory for Neural Computation and Adaptation, RIKEN Center for Brain Science, 2-1 Hirosawa, Wako, Saitama 351-0198, Japan; shun.ogawa@riken.jp

[3]  Space Research Institute, Profsoyuznaya 84/32, 117997 Moscow, Russia; valex@iki.rssi.ru

[4]  Aix Marseille Univ., Université de Toulon, CNRS, CPT, Marseille, France

\*  Correspondence: xavier.leoncini@cpt.univ-mrs.fr

**Abstract:** Starting from the given passive particle equilibrium particle cylindrical profiles, we built self-consistent stationary conditions of the Maxwell-Vlasov equation at thermodynamic equilibrium with non-flat density profiles. The solutions to the obtained equations are then discussed. It appears that the presence of an azimuthal (poloidal) flow in the plasma can ensure radial confinement, while the presence of a longitudinal (toroidal) flow can enhance greatly the confinement. Moreover in the global physically reasonable situation, we find that no unstable point can emerge in the effective integrable Hamiltonian of the individual particles, hinting at some stability of the confinement when considering a toroidal geometry in the large aspect ratio limit.

**Keywords:** Vlasov equation; hot plasma; Hamiltonian dynamics; magnetized fusion

## 1. Introduction

Insuring confinement of a hot plasma using a magnetic field, is one of the key issues to sustain in order to achieve magnetically confined fusion reactors. In these regards the emergence of a transport barrier that gives rise to the so-called H-mode has been a key ingredient in the design of most recent machines [1,2]. The study of these barriers has generated much work in the literature [3–5], but even though observed experimentally, up to our knowledge, there are currently no clear theoretical explanations of the emergence of such a barrier, neither has it really been observed as emerging from self-consistent numerical simulations (without some especially tailored external forcing). Following our recent study on the equilibrium solutions of passive particles described in [6], we built full self-consistent equilibrium solutions of the classical Maxwell-Vlasov equations. We then discuss what kind of motion can be expected in this self-consistent setting regarding the integrable motion of passive charged particles. We find out that in this setting no hyperbolic point emerges when looking at the effective potential driving the motion, meaning that in these type of cylindrical configurations, we do not expect a breaking of the magnetic moment when going back to a toroidal geometry due to separatrix crossings such as the one displayed in [7] or the presence of Hamiltonian chaos due to this mechanism [7–11]. However since we have a full family of solutions, we can as well study the influence of the presence of some "toroidal" flow in the plasma, on the plasma density profile, and we show it can enhance plasma confinement. Our approach stems from first principles, so it is considered in a very idealized situation, but we believe it could be useful and shed possible light on the origin of improved plasma confinement, such as the rise of transport barriers and how to trigger

them. This paper is organized as follows: in the first part we present the idealized configuration and the passive particle thermal equilibrium from [6], then building from this we discuss the self-consistent approach with one species, and finally move on to a real full self-consistent solution with two species.

## 2. Plasma Setting and Passive Equilibrium

Let us recall the conditions that were discussed in [6]. We consider an infinite aspect ratio limit of the tokamak torus such that we end up with a cylindrical configuration. In this configuration, we choose an ideal magnetic field of the form

$$\overrightarrow{B(r)} = B_0 \hat{e}_z + B_0 g(r) \hat{e}_\theta \,,$$

whose vector potential field in a Coulomb gauge writes.

$$\overrightarrow{A(r)} = \frac{B_0 r}{2} \hat{e}_\theta - B_0 F(r) \hat{e}_z \,,$$

where $\hat{e}_z$ is the unit vector along the cylinder axis (corresponding to the toroidal direction in the torus), $\hat{e}_\theta$ the unit vector on the poloidal direction, $B_0$ is the intensity of the uniform magnetic field (assumed to be generated by external coils), and $F$ is the primitive of $g$, that can be linked to the plasma current $j_z(r)$ or the so-called $q$-profile (introducing some fictitious characteristic radius $R_{\text{per}}$) by

$$F(r) = \int^r \frac{r dr}{R_{\text{per}} q(r)} = \int^r \frac{j_z(r) r}{B_0} dr \,.$$

We shall consider a positive charged particle evolving in such magnetic filed, without any electric field. The motion of the particle is then described by the following effective one degree of freedom Hamiltonian

$$H = \frac{1}{2m} \left[ p_r^2 + \left( \frac{p_\theta}{r} - \frac{q B_0 r}{2} \right)^2 + (p_z - q A_z(r))^2 \right] \,. \tag{1}$$

Indeed due to the symmetries of the problem (translation along $z$ and rotation of $\theta$ around the cylinder axis), we have on top of the kinetic energy of the system (the Hamiltonian $H$.) two extra conserved quantities $p_\theta$ and $p_z$, turning this 3-degree of freedom problem into an integrable one with an effective one-degree of freedom Hamiltonian. Here we have an effective potential energy that depends on the initial condition through the constant values of $p_\theta$ and $p_z$. We note as well that the effective potential depends on the poloidal magnetic field through the function $F(r)$, it is then possible to imagine settings that give rise to a hyperbolic point and an associated separatrix. The presence of these separatrices in the full phase space can then after some perturbation be the natural root of separatrix chaos and the breaking of adiabatic invariants through separatrix crossings. We shall come back to this statement, as this feature was shown to break the magnetic moment in some regions of the full phase space [7].

Let us now built a passive particle equilibrium distribution from the previous considerations. Using the Hamiltonian (1), we can find a stationary one particle-density function obeying some Vlasov equation [12–15] we just need to find $f(p_r, p_\theta, p_z, r, \theta, z, t)$ such that

$$\frac{Df}{Dt} = 0 \,,$$

where we used the particle derivative with the specificity that

$$\frac{\partial f}{\partial t} = 0 \,.$$

Solutions of this problem satisfy necessarily the following equation

$$\{f, H\} = 0 \,,$$

where $\{\cdot, \cdot\}$ denotes the Poisson brackets. Any function $f$ that is just a function of $H$ is a solution of the problem. To choose one among the infinite possibility, we applied the maximum entropy principle [16–18], imposing constraints due to the different invariants of the dynamics coming from the symmetries of the problem and the total number of particles in the system. These lead to the introduction of three Lagrangian multipliers $\beta$, $\gamma_\theta$, $\gamma_z$ respectively associated with the energy, the angular momentum and the translational invariance. We then obtain as a result the following density function,

$$f \propto e^{-\beta H - \gamma_\theta p_\theta - \gamma_z p_z} \,, \tag{2}$$

$\gamma_1$ the last multiplier dealing with the normalization of $f$ (i.e., conservation of number of particles $N$, $\int f d^3\mathbf{p} d^3\mathbf{q} = N$), is absorbed in the proportional coefficient of (2).

From this stationary distribution we can get for instance the spatial density $n(\mathbf{q})$ as (see Appendix A for more details)

$$n(\mathbf{q}) \equiv \int f_0 d^3\mathbf{p} = \int f_0 r^{-1} dp_\theta dp_z dp_r. \tag{3}$$

We get equivalently the charge density by

$$\rho(r) = \frac{Nq \exp\left(-ar^2 - bA_z(r)\right)}{4\pi^2 R_{\text{per}} \int_0^\infty r \exp\left(-ar^2 - bF(r)\right) dr}, \tag{4}$$

where

$$a = \frac{\gamma_\theta}{2}\left(qB_0 - \frac{m\gamma_\theta}{\beta}\right), \quad b = \gamma_z qB_0.$$

different situations may arise in this profile and we refer to [6] for some of the results and conclusions. Let us insist on the fact that the Lagrangian multipliers $\gamma_z$ is due to the conservation of $p_z$, and is directly linked to its average. A non zero $\gamma_z$ corresponds therefore to the existence of some global movement of the plasma, and mass flow. Of course, as shall be discussed later, motion of ions induces as well a current, but it is important to recall that if there is no global flow ($\gamma_z = 0$ and $\gamma_\theta = 0$), then the equilibrium is trivial and there is just a flat profile, and non-flat profiles in an non flowing plasma correspond then to an out of equilibrium states. However, now let us move to the construction of a self-consistent solution of the problem.

## 3. Self-Consistent Equation

### 3.1. One Species in Neutralizing Background

Before moving to self-consistency, we will be more specific in what is meant by self-consistent. We consider that the electric and magnetic fields are created by the particles, using Gauss and Ampére laws, meaning we assume that those fields are static and created by the stationary distribution, we as well consider non-relativistic particles. To build on what has been done before, we consider first a system with one species of charged particles and without electric field. In other words there is some static background that neutralizes the charge density. Moreover we will consider that the component of the magnetic field along the cylinder axis $B_0\hat{e}_z$ is generated by some external coils.

If one now looks back at the passive particle distribution one notice that it is possible as well to compute the density of current flowing along the axis by performing the integration

$$j_z(r) = q \int v_z f_0 d^3 p,$$

where

$$v_z = \frac{p_z - qA_z(r)}{m},$$

and we obtain

$$
\begin{aligned}
j_z(r) &= \frac{-Nq\gamma_z \exp\left(-ar^2 - bA_z(r)\right)}{4\pi^2 R_{\text{per}}\beta \int_0^\infty r \exp\left(-ar^2 - bF(r)\right) dr} \\
&= -\frac{\gamma_z}{\beta}\rho(r)
\end{aligned}
\tag{5}
$$

To move to a partial self-consistent setting, we neglect the influence of the poloidal current on the axial component of the magnetic field which will remain $B_0\hat{e}_z$. This can be for instance justified when $B_0$ is large and plasma density low, we may consider for instance when $qB_0 \gg m\gamma_\theta/\beta$) such that $a$ is linear in $\gamma_\theta$, however having in mind an analogy with the tokamak, the poloidal magnetic field is generated by the current of the particles, we shall have

$$\Delta A_z = -\mu_0 j_z(r),\tag{6}$$

so we end up with an implicit equation in the vector potential which if it has solutions leads to self-consistent solutions of the Vlasov system, that explicitly writes

$$\frac{1}{r}\frac{d}{dr}\left(r\frac{dA_z(r)}{dr}\right) \propto \gamma_z e^{-ar^2 - bA_z(r)}.$$

Removing all the dimensions and multiplicative terms by rescaling, we end up with an equation,

$$\frac{1}{\bar{r}}\frac{d}{d\bar{r}}\left(\bar{r}\frac{d\psi(\bar{r})}{d\bar{r}}\right) = -e^{-\bar{a}\bar{r}^2 + \psi(\bar{r})},\tag{7}$$

where we introduced variable $\bar{r}$ and the function $\psi(\bar{r})$. For simplicity we now will get rid of the $\bar{\phantom{r}}$ term in the following. We can notice that, when making the change of variables in the end the sign of $\gamma_z$ becomes irrelevant as it introduces a dependence on $\gamma_z^2$. This is not really surprising as the end result should not depend on how the direction of the plasma current flows, for the sake of simplicity we will assume in the rest of the paper that $\gamma_z > 0$.

### 3.2. Full Self-Consistent Solution

To find a full self-consistent solution of the Vlasov system, we do not assume a neutralizing background, in order to still be coherent we then need to add a second type of particles whose charges have an opposite charge to the initial ones (for instance electrons if we had protons or alpha particles). Since we have a choice, we shall consider that the previous equilibrium was computed for positive charges ($q > 0$). Then for the neutralizing negative particles, we will follow the same procedure and maximize each entropy of each type of particles independently (this hypothesis could be discussed, in someway we are assuming no collisions, even if we will need to impose electroneutrality). We will use the $\pm$ sign to refer for instance to the distribution of positive and negative particles, so the previous expressions, for instance $\rho(r)$ from Equation (4) will be noted $\rho^+$(and multiplied by the charge of the particle to become the positive charge density), and the same for the current $j_z$ from (5) will become $j_z^+$. So lets us compute $f^-, \rho^-(r), j_z^-(r)$, the Hamiltonian of the passive negative particles writes

$$H^- = \frac{(p_r^-)^2}{2m^-} + \frac{\left(\frac{p_\theta^-}{r} + \frac{qB_0 r}{2}\right)^2}{2m^-} + \frac{(p_z^- + qA_z(r))^2}{2m^-},$$

which lead to the passive distribution

$$f^- \propto e^{-\beta^- H^- - \gamma_\theta^- p_\theta^- - \gamma_z^- p_z^-}$$

and density of negative charges

$$\rho^-(r) = \frac{-q e^{-a^- r^2 - b^- A_z(r)}}{4\pi^2 R_{\text{per}} \int r e^{-a^- r^2 - b^- A_z(r)} dr}$$

with $a^- = -\frac{\gamma_\theta^-}{2}\left(qB_0 + \frac{m^- \gamma_\theta^-}{\beta^-}\right)$, $b^- = -q\gamma_z^-$. We can as well in order to move to self consistency, compute the current induced by these negative charges

$$j_z^-(r) = \frac{q\gamma_z^- e^{-a^- r^2 - b^- A_z(r)}}{4\pi^2 R_{\text{per}} \beta^- \int r e^{-a^- r^2 - b^- A_z(r)} dr},$$

And we end up with the self-consistent Poisson type equation that writes:

$$\frac{1}{r}\frac{d}{dr}\left(r\frac{dA_z(r)}{dr}\right) = -j_z^+(r) - j_z^-(r)$$

and since we assumed no electric field, we as well have to impose electro-neutrality:

$$\rho^-(r) + \rho^+(r) = 0, \tag{8}$$

which leads to

$$- C_- e^{-a^- r^2 - b^- A_z(r)} + C_+ e^{-a^+ r^2 - b^+ A_z(r)} = 0 \tag{9}$$

where

$$C_\pm = \frac{Nq}{4\pi^2 R_{\text{per}} \beta^\pm \int r e^{-a^\pm r^2 - b^\pm A_z(r)} dr}.$$

Let us now discuss the possible solutions.

- The first case to consider is if $b^- \neq b^+$, Equation (9) then implies that $A_z$ can eventually depend on $r^2$

$$A_z(r) = \frac{\log\left(\frac{C_-}{C_+}\right)}{(b^- - b^+)} + \left(\frac{a^+ - a^-}{b^- - b^+}\right) r^2$$

  and that we end up with a uniform constant current $j_z$. This is not really interesting physically.

- The other case of possible non-trivial solution which satisfy self consistency and electroneutrality implies $b^- = b^+ = b$ and therefore $a^- = a^+ = a$ and $C_+ = C_- = C$, this leads to a non-uniform current

$$j_z(r) = -Cq\gamma_z^+ e^{-ar^2 - bA_z(r)}\left(\frac{1}{\beta^+} + \frac{1}{\beta^-}\right)$$

  with possibly the two species at different temperatures. The self-consistent equations then ends up writing as

$$\frac{1}{r}\frac{d}{dr}\left(r\frac{dA_z(r)}{dr}\right) = Cq\gamma_z^+ e^{-ar^2 - bA_z(r)}\left(\frac{1}{\beta^+} + \frac{1}{\beta^-}\right)$$

  In addition, if we get rid of the dimensions, we end up with a formally identical Equation as (7),

$$\frac{1}{\bar{r}}\frac{d}{d\bar{r}}\left(\bar{r}\frac{d\psi(\bar{r})}{d\bar{r}}\right) = -e^{-\bar{a}\bar{r}^2 + \psi(\bar{r})}.$$

In this context adding a second species ends up solving the same implicit equation, but we where then able to get rid of the neutralizing background assumption and potentially be more physically relevant.

## 4. Solutions of the Self-Consistent Equation

To solve Equation (7) we can envision different situations, namely $a > 0$, $a = 0$ and eventually $a < 0$. To tackle the problem we can as well make some transformations, if we note $\phi(r) = -ar^2 + \psi(r)$, we end up with

$$\Delta\phi(r) + 4a = -e^{\phi(r)}, \tag{10}$$

but it is not much simpler. We now assume that $a > 0$, so we will have to deal with a $\pm$ sign if the initial $a \neq 0$. We can then rescale the length again and obtain

$$\Delta\phi(r) \pm 1 = -e^{\phi(r) - \log(4a)}, \tag{11}$$

and then shift $\phi$ to arrive at

$$\Delta\phi(r) \pm 1 = -e^{\phi(r)}. \tag{12}$$

Since $r > 0$, we may use the notation $t = \log r$, and we end up with

$$\ddot{\phi} = -e^{\phi + 2t} \mp e^{2t},$$

where the ˙ refers to $d/dt$ and if we note $\varphi = \phi + 2t$ (in some way we change our reference frame), we finally obtain

$$\ddot{\varphi} = -e^{\varphi} \mp e^{2t}, \tag{13}$$

where we can recognize some newton equation, with a force deriving from a potential and one that is just time dependent (when $a = 0$ the time dependent force is simply zero and we have an autonomous system).

### 4.1. Bennett-Pinch Solution When $a = 0$

A family of solutions of this equation can be found in the literature under the name of Bennet pinch solution [19,20], it occurs when we consider a distribution with $a = 0$. In fact, the Bennett pinch corresponds to a particular case of the general solutions of the Lane Emden-like equation when for $a = 0$ which means $\gamma_\theta = 0$ or $\gamma_\theta = \frac{qB_0\beta}{m}$.

$$\Delta\psi(r) = -e^{\psi(r)}, \tag{14}$$

which corresponds to the use of any holomorphic function $w$ as

$$e^{\psi(r)} = \frac{8|w'|^2}{(1 + |w|^2)^2}, \quad \text{where} \quad w' = \frac{dw}{dz}. \tag{15}$$

The function $w$ should holds that both $|w|$ and $|w'|$ depend only on $r = |z|$. For more details a good starting point can be found in [21]. A trivial example is that $w = cr^\alpha$. In this case,

$$e^{\psi(r)} = \frac{8c^2\alpha^2 r^{2\alpha - 2}}{(1 + c^2 r^{2\alpha})^2}. \tag{16}$$

and the Bennet pinch solution corresponds to the choice $\alpha = 1$:

$$e^{\psi(r)} = \frac{8c^2}{(1 + c^2 r^2)^2}. \tag{17}$$

We may also check a bit the newton equation (13) in this case, we end up with a conservative system with a force deriving from a potential in one dimension, so we can solve it. We can define a Hamiltonian

$$H = \frac{p^2}{2} + e^q , \tag{18}$$

and find the constant energy trajectories. Given the shape of the potential energy, we see that for a given total energy $E$ (that is always positive), we will have a maximum of $q$ attained when the momentum is $p = 0$ at $q^* = \log E$, and when $q \to -\infty$, we will end up with momenta $p_\infty = \pm\sqrt{E}$, so we can sketch a phase portrait as depicted in Figure 1, and can find again the solutions analytically using the first order ordinary differential equation given by the Hamiltonian (18).

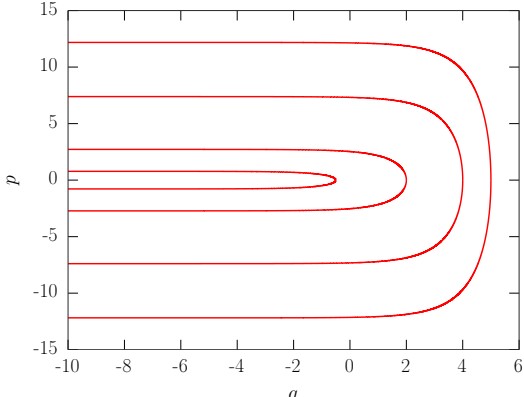

**Figure 1.** Phase portrait of Hamiltonian (18).

### 4.2. Solution for $a \neq 0$

We had left with

$$\ddot{\varphi} = -e^\varphi - e^{2t} , \tag{19}$$

We can directly obtain a Hamiltonian

$$H = \frac{p^2}{2} + e^q \pm q e^{2t} . \tag{20}$$

but we may also $\delta\varphi = \varphi - 2t$, we end up with

$$\ddot{\delta\varphi} = -e^{\delta\varphi + 2t} - e^{2t}$$
$$= -e^{2t}(1 + e^{\delta\varphi})$$

So we can summarize the dynamics of the system with a different Hamiltonian of the type

$$H = \frac{p^2}{2} + e^{2t}(e^q \pm q) . \tag{21}$$

Unfortunately, this is a time-dependent Hamiltonian, and as such not easy to resolve. Similar Hamiltonian with an exponential growth of the amplitude of the potential appears in the literature, such as for instance in systems dealing with wave particle interactions [22,23]; however the techniques applied there do not seem to be directly applicable, so in order to look at the behavior of the potential we need to resort to a numerical resolution.

### 4.3. Numerical Solutions for $a \neq 0$

Looking at Equation (12), we may define an adimensionalized current $j$ as $j = \exp(\phi)$, since we expect $j$ to be smooth at the origin $\frac{dj}{dr}(0) = 0$, this implies as well that $\frac{d\phi}{dr}(0) = 0$. We shall thus impose this in our initial conditions.

We may anticipate that solutions with $a < 0$ ($-$ sign in Equation (12) or $a = -1/4$ in Equation (7)) are not what we would expect from a physical behavior. We can as well notice that $\phi = 0$ is a solution (fixed point) that implies as well $j = 1$, so a uniform current. Looking at the numerical solution, we can see in Figure 2 that the fixed point is actually an attractor and that we observe current oscillations. The full solution is indeed not realistic at least when $r \to \infty$, but it could have some meaning for small values of $r$.

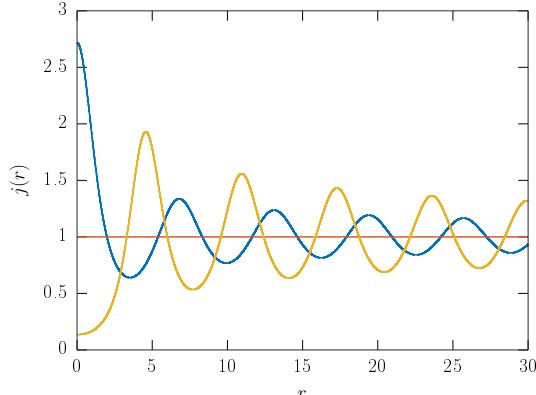

**Figure 2.** Solutions of Equation (12) with the minus sign. The fixed point $j = 1$ is an attractor and some current oscillations are observed. Initial conditions are $\phi(0) = 1$ (blue), $\phi(0) = 0$ (orange) and $\phi(0) = -2$ (yellow).

Let us now turn to more relevant physical solutions with $a > 0$ ($+$ sign in Equation (12) or $a = 1/4$ in Equation (7). Solutions are displayed in Figure 3. We notice that they are quite regular and no special behavior is observed when the initial condition is changed, we can expect some flattening when $\phi(0)$ becomes small. We can from this derive some behavior of $\phi$ assuming $\phi(0) \ll 0$, we can then neglect the exponential term in Equation (12)), which leads to $\phi(r) \approx -r^2/4 + \phi(0)$, and we see this implies that $\psi(r)$ becomes some constant, and we can thus explain the flattening of the current profile.

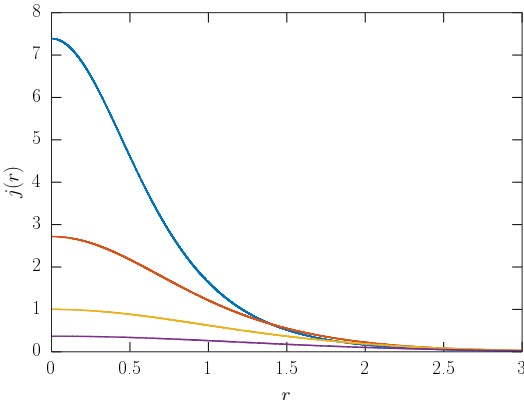

**Figure 3.** Solutions of Equation (12) with the plus sign. All profile are monotonous and decrease quite fast and the fixed point $j = 0$ is an attractor. Initial conditions are $\phi(0) = 2$ (blue), $\phi(0) = 1$ (orange), $\phi(0) = 0$ (yellow) and $\phi(0) = -1$ (magenta).

Since the current profile is directly proportional to the density profile of the plasma, we can see that as soon as $\phi(0) \geq 0$ that the presence of a plasma flow along the $z-$axis enhances plasma confinement, this is illustrated in Figure 4.

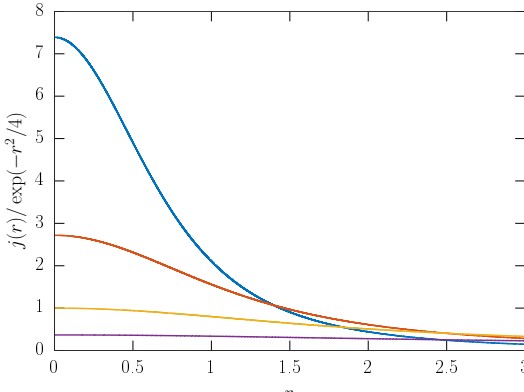

**Figure 4.** Influence of the presence of the non-zero $\gamma_z$ in the profile. We can see that as soon as $\phi(0) > 0$, we end up with a better radial confinement of the plasma. We consider the solutions of Equation (12) with the plus sign. Initial conditions are $\phi(0) = 2$ (blue), $\phi(0) = 1$(orange), $\phi(0) = 0$ (yellow) and $\phi(0) = -1$ (magenta).

When looking for unstable points and separatrices in in the phase portrait of the integrable Hamiltonian (1), with a self-consistent component of the magnetic field, we found out that the solutions with $a > 0$ are not able to create a strong enough modulation to trigger the presence of an unstable point. Given the conclusions found in [7], we thus expect that these configurations should lead as well to regular trajectories when going to a toroidal configuration, at least in the large aspect ratio limit and this hints towards some stability of these solutions. On the contrary when $a \leq 0$ it is possible to find self-consistent configurations that lead to the presence of a separatrix, and thus radial chaotic motion of particles can be expected if such configurations locally appear in the plasma, the phenomena are illustrated in Figure 5.

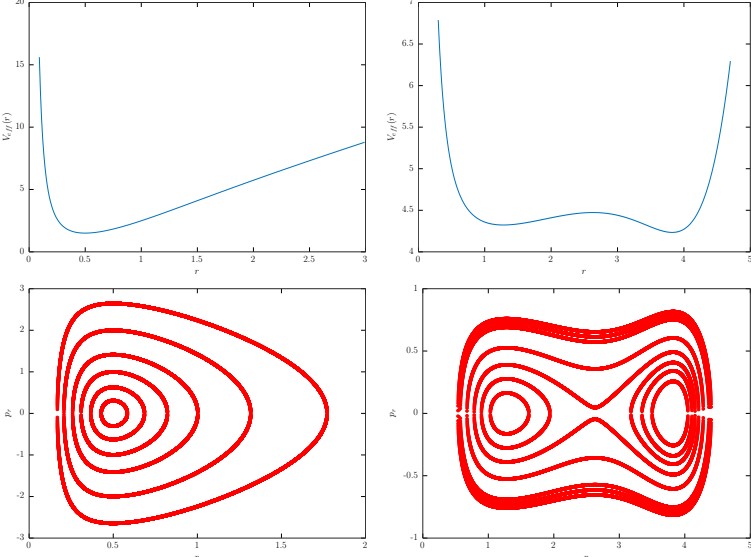

**Figure 5.** Illustration of the effective potential (**top**) and resulting phase space trajectories (**botom**) arising with a self-consistent magnetic field. Left: A typical situation when $a > 0$ (here $a = 1/4$), and $\phi(0) = 2$; the effective potential is computed with $p_\theta = 0.5$ and $p_z = 3$. Right: A situation when $a < 0$ (here $a = -1/4$) and $\phi(0) = -2$, and unstable arises; the effective potential is computed with $p_\theta = 0.5$ and $p_z = -4$.

## 5. Conclusions

In this paper, we built self-consistent solutions of the Vlasov equation in a cylindrical magnetized plasma, these solutions correspond to a thermodynamic equilibrium of a non self-consistent "passive" plasma, and as such can potentially accommodate with a different thermodynamic temperatures of the ions and the electrons. Besides the possibility of using these exact solutions as test beds for large numerical kinetic simulations and code validation, such as for instance GYSELA [24], we found out that these solutions confirm the results proposed in [6], and that a better radial confinement of the plasma can be obtained by the existence of a longitudinal flow (i.e., a toroidal flow in a toroidal machine) above a certain threshold. It could be therefore interesting to check whether there is an experimental correlation of these findings, namely if a better confinement of the plasma in magnetized fusion devices can not be obtained by generating a toroidal momentum of the plasma while heating it with neutral injections beams. Finally, our findings show that equilibrium configurations leading to a globally confined plasma ($a > 0$) are unable to create unstable points and associated separatrices in the radial direction, which hints at some stability of these magnetic profiles when considering a confinement within a torus and no destruction of the adiabatic invariant, at least in the large aspect ratio limit, and as long as electric field effects can be neglected, and this is thus compatible with results discussed in [25] regarding special Bennett type solutions. As a perspective of this work, we may now consider taking into account global diamagnetic effects and see the possible influence of poloidal flow and current on the confinement. This appears as a crucial point, as when we go back to the toroidal geometry, the destruction of one symmetry will inevitably imply a coupling between both toroidal and poloidal flows of the plasma. So the influence of a poloidal flow on confinement needs as well to be addressed in a self-consistent manner and some comparison with results obtained in the MHD framework [26,27] could be as well envisioned.

**Author Contributions:** Investigation, E.L., S.O., G.D.-P., A.V., X.G. and X.L.

**Funding:** This work has been carried out within the framework of the French Research Federation for Magnetic Fusion Studies. The project leading to this publication (TOP project) has received funding from Excellence Initiative of Aix-Marseille University—A*MIDEX, a French "Investissements d'Avenir" programme.

**Conflicts of Interest:** The authors declare no conflict of interest.

## Appendix A. Density and Current Computation

Once the particle density function is obtained, in order to compute quantities such as the particle density or the current we rewrite $-\beta H - \gamma_z p_z - \gamma_\theta p_\theta - \gamma_0$ (with $q = m = 1$) as

$$
-\frac{\beta}{2}\left[ p_r^2 + \left( \frac{p_\theta}{r} - \left( \frac{B_0}{2} - \frac{\gamma_\theta}{\beta} \right) r \right)^2 + \left( p_z - \left( A_z(r) - \frac{\gamma_z}{\beta} \right) \right)^2 \right]
$$

$$
-\frac{\gamma_\theta}{2}(B_0 - \frac{\gamma_\theta}{\beta})r^2 - \gamma_z A_z(r) + \frac{\gamma_z^2}{2\beta} - \gamma_0 \,, \quad \text{(A1)}
$$

and then just make some Gaussian integrals.

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
