# Peer review of "Influence of Toroidal Flow on Stationary Density of Collisionless Plasmas"

_fluids, doi:10.3390/fluids4030172_

Round 1

Reviewer 1 Report

This is an interesting and clearly written article on a subject that has been frequently investigated in the literature in a number of contexts.

I would like to raise three points

1) I am confused by the use of the word flow (poloidal, toroidal flows). Actually currents are discussed, not mass flows. I understand that by properly combining the different species currents one can derive the plasma mass flow, but I do not see it done in this paper.                                Should the word flow be simply replaces by currents in the text?

2) Is it fully consistent to study the effect of a (in particular of a < 0) since B_z is taken to be not affected by the poloidal currents?

3) The method employed in eq 15 goes back to Poincare’ (see e.g. Two-dimensional Harris-Liouville plasma kinetic equilibria, F. Ceccherini, et al/ Phys. Plasmas, 12, 052506 (2005) but it has been used in many other works)

Author Response

We first thank the referee for his constructive remarks. We feel that the revised paper and the results presented in it have gained in clarity and are more prone to avoid misunderstandings. We do hope that this revised manuscript will meet the criteria needed for publication in Fluids  

This is an interesting and clearly written article on a subject that has been frequently investigated in the literature in a number of contexts. I would like to raise three points I am confused by the use of the word flow (poloidal, toroidal flows). Actually currents are discussed, not mass flows. I understand that by properly combining the different species currents one can derive the plasma mass flow, but I do not see it done in this paper. Should the word flow be simply replaces by currents in the text?

We have slightly modified the text in order to be more clear. The flow/current of each species is governed by their respective γ z which are more or less equal in order to insure electroneutrality. Since the mass of the electron is much smaller than the one of the ions (which are assumed to be light species), this results in a global flow along the z direction of the plasma

Is it fully consistent to study the effect of a (in particular of a < 0) since B_z is taken to be not affected by the poloidal currents?

We are currently considering the full self-consistent profile, as here we neglected the poloidal current, assuming that the magnetic field generated by the coils is much larger than the self-generated one. It is then possible to consider that solutions taking into account the effects of a , can give some hints, finding similar profile, could be for instance linked to a “local” equilibrium. We have added a comment in the paper to be more explicit.

The method employed in eq 15 goes back to Poincare’ (see e.g. Two-dimensional Harris-Liouville plasma kinetic equilibria, F. Ceccherini, et al/ Phys. Plasmas, 12, 052506 (2005) but it has been used in many other works)

We thank the referee for pointing us to this reference, which is now included in the paper.

Reviewer 2 Report

Refereee report.
Title: "Influence of toroidal flow on stationary density of collisionless plasmas"
Authors: E. Laribi, S. Ogawa, G. Dif-Pradalier, A. Vasiliev, X. Garbet, X. Leoncini

In the manuscript the authors study families of stationary, thermal-equilibrium, solutions of the system of the Maxwell-Vlasov equations, describing a plasma in a cylindrical geometry. First, the authors discuss a "passive" equilibrium, where a two-component (axial and azimuthal) magnetic field is prescribed with null electric field, derived by analyzing single-particle dynamics. Next, a (quasi) self-consistent stationary solution is considered in an electro-neutral configuration. The authors discuss the derivation and describe the properties of this solution.
In particular, they found that the presence of an axial flow tends to increase the
plasma confinement, in that the density profile tends to peak at the cylinder axis.
The paper is clearly written and the derivation is quite detailed; implications of the
results on the confinement of a toroidal plasma are discussed.
However, in my opinion, there are some point (which I indicate in the following) that the authors should address before publication. This could help to improve the clarity of the paper.

1) Concerning the density function expressed by equation (2): it seems to me that p_\theta and p_z can have both positive and negative values. Therefore, f exponentially diverges in the limit where p_theta and/or p_z have large negative values.
In principle, this divergence could lead to a problem when integrating f with respect to p_theta or p_z (e.g., equation (3)). Could the author be more specific on that point, for instance giving more details in the derivation of expressions (4) and (5) ?

2) On page 4 (just after eq. (5)) it is stated that the influence of the poloidal current j_theta on the axial magnetic field is neglected, so that Bz is somehow "forced" to be uniform, regardless of the form of j_theta (which is actually not calculated). This assumption, which in principle can affect the self-consistency of the solution, is not sufficiently discussed. For instance: which is the situation where this approximation can be considered as valid? (large B0)? To which extent a modulation
of Bz induced by j_theta can affect the results?

3) On page 8 the authors describe the derivation of numerical solutions of equation (13). In this equation the independent variable is represented by t, which is related to the radial variable r by the relation t = log (r). Solutions are found in a spatial domain where r is in a range between r=0 and r=r_max (see Fig.s 2, 3, 4). Since the value r=0 corresponds to t=-infinity, an infinitely wide spatial domain is to be considered when the variable t is used. Can the authors give more details on the numerical solution, illustrating how they manage this infinite range? How do they impose the initial condition at r=0 (corresponding to t=-infinity)?

4) On page 8 the authors mention the phase portrait of Hamiltonian (1) in the case of selfconsistent solutions (corresponding to both to a>0 and to a<0), and the possible presence of separatrices and unstable points. Can they give some more details on that interesting point, for instance by illustrating such a phase portrait in a figure, for both cases?

Minor points:
a) In some equations (e.g., eq. (2)) proportionality is indicated by the symbol "~". Since such a symbol
commonly means "about equal", I suggest to replace it by the proper "proportional to" symbol.

b) Captions of Fig.s 3 and 4: which colour of the lines corresponds to which initial condition?

c) There are some misprints in the text. Moreover, the english style could be somehow improved.

Author Response

We first thank the referee for his constructive remarks. We feel that the revised paper and the results presented in it have gained in clarity and are more prone to avoid misunderstandings. We do hope that this revised manuscript will meet the criteria needed for publication in Fluids

In the manuscript the authors study families of stationary, thermal-equilibrium, solutions of the system of the Maxwell-Vlasov equations, describing a plasma in a cylindrical geometry. First, the authors discuss a "passive" equilibrium, where a two-component (axial and azimuthal) magnetic field is prescribed with null electric field, derived by analyzing single-particle dynamics. Next, a (quasi) self-consistent stationary solution is considered in an electro-neutral configuration. The authors discuss the derivation and describe the properties of this solution.

In particular, they found that the presence of an axial flow tends to increase the plasma confinement, in that the density profile tends to peak at the cylinder axis. The paper is clearly written and the derivation is quite detailed; implications of the results on the confinement of a toroidal plasma are discussed.

However, in my opinion, there are some point (which I indicate in the following) that the authors should address before publication. This could help to improve the clarity of the paper.

Concerning the density function expressed by equation (2): it seems to me that p_\theta and p_z can have both positive and negative values. Therefore, f exponentially diverges in the limit where p_theta and/or p_z have large negative values. In principle, this divergence could lead to a problem when integrating f with respect to p_theta or p_z (e.g., equation (3)). Could the author be more specific on that point, for instance giving more details in the derivation of expressions (4) and (5) ?

We have added the idea of the derivation of the Gaussian integration in the appendix, as can be seen, the quadratic term for both p z and p θ avoids any problems in the integration.

On page 4 (just after eq. (5)) it is stated that the influence of the poloidal current j_theta on the axial magnetic field is neglected, so that Bz is somehow "forced" to be uniform, regardless of the form of j_theta (which is actually not calculated). This assumption, which in principle can affect the self-consistency of the solution, is not sufficiently discussed. For instance: which is the situation where this approximation can be considered as valid? (large B0)? To which extent a modulation of Bz induced by j_theta can affect the results?

Indeed we assume that B 0 is large, so that the longitudinal component of the magnetic field is not much changed by the plasma self-generated one. We are currently investigating a full self-consistent calculation, from our preliminary results, the self-consistent equations give rise to two coupled ordinary differential equations, and a full study of the behavior becomes more complex. We added a sentence in the paper about this remark and insisted on the large B 0 approximation.

On page 8 the authors describe the derivation of numerical solutions of equation (13). In this equation the independent variable is represented by t, which is related to the radial variable r by the relation t = log (r). Solutions are found in a spatial domain where r is in a range between r=0 and r=r_max (see Fig.s 2, 3, 4). Since the value r=0 corresponds to t=-infinity, an infinitely wide spatial domain is to be considered when the variable t is used. Can the authors give more details on the numerical solution, illustrating how they manage this infinite range? How do they impose the initial condition at r=0 (corresponding to t=-infinity)?

The change of variables is actually only made from the theoretical standpoint, numerically, we did not make this change of variable and kept r . The initial condition, is then fixed by the value of φ ( 0 ) , and we imposed that φ '( 0 ) =0 , and then use a simple integrator lsode provided by octave software and rewrote the second order equation, into a one order one using the variables X=( φ ( r ) , φ '( r ) ) .

On page 8 the authors mention the phase portrait of Hamiltonian (1) in the case of selfconsistent solutions (corresponding to both to a>0 and to a<0), and the possible presence of separatrices and unstable points. Can they give some more details on that interesting point, for instance by illustrating such a phase portrait in a figure, for both cases?

We have added a figure to illustrate the point regarding separatrices.

Minor points: In some equations (e.g., eq. (2)) proportionality is indicated by the symbol "~". Since such a symbol commonly means "about equal", I suggest to replace it by the proper "proportional to" symbol.

We have made the changes.

Captions of Fig.s 3 and 4: which colour of the lines corresponds to which initial condition?

We have added to which colour each line corresponds.

There are some misprints in the text. Moreover, the english style could be somehow improved.

We have corrected indeed numerous mistakes, we hope the style is overall improved.

Round 2

Reviewer 1 Report

The points that had been raised by the referees have been answered satisfactorily